# Neutrophile-to-Lymphocyte Ratio as a Predictor of Mortality and Response to Treatment in Invasive Aspergillosis among Heart Transplant Recipients—Exploratory Study

**DOI:** 10.3390/medicina57121300

**Published:** 2021-11-27

**Authors:** Tomasz Urbanowicz, Michał Michalak, Anna Olasińska-Wiśniewska, Bartłomiej Perek, Michał Rodzki, Hanna Wachowiak-Baszyńska, Marek Jemielity

**Affiliations:** 1Cardiac Surgery and Transplantology Department, Poznan University of Medical Sciences, 61-848 Poznan, Poland; anna.olasinska@poczta.onet.pl (A.O.-W.); bperek@ump.edu.pl (B.P.); michal.rodzki@skpp.edu.pl (M.R.); hwbaszynska@gmail.com (H.W.-B.); mjemielity@poczta.onet.pl (M.J.); 2Department of Computer Science and Statistics, Poznan University of Medical Sciences, 60-808 Poznan, Poland; michal@ump.edu.pl

**Keywords:** aspergillosis, htx, transplantation, NLR, IA

## Abstract

*Background and objective:* Aspergillus pulmonary infections are potentially life-threatening complications that can occur after heart transplantation. The aim of the study was to find an easily available mortality predictor during Aspergillosis infection therapy following heart transplantation. *Materials and methods:* This study involved 15 heart recipients with the mean age of 55 ± 6 years who were diagnosed with invasive aspergillosis (IA) in a mean time of 80 ± 53 (19–209) days after orthotropic heart transplantation. *Results:* Out of fifteen patients diagnosed with IA, five died. The mean time from diagnosis to death in the deceased group was 28 ± 18 days. They were diagnosed with IA in a mean time of 80 ± 53 (19–209) days after orthotropic heart transplantation. During the initial seven days of therapy, the neutrophil to lymphocyte ratio (NLR) significantly differed between the two groups on day three and day seven, with median values of 10.8 [4.3–17.0] vs. 20.2 [17.4–116.8] (*p* = 0.0373) and 5.2 [3.2–8.1] vs. 32.2 [13.5–49.9] (*p* = 0.0101) in the survivor and the deceased group, respectively. The NLR was a significant predictor of death both on day three (cut-off point 17.2) and day seven (cut-off point 12.08) of therapy. *Conclusions:* Findings in our study indicate that NLR may be of predictive value in the estimation of mortality risk or response to treatment among patients with invasive aspergillosis following heart transplantation.

## 1. Introduction

Pulmonary infections are common and potentially life-threatening complications that can occur after heart transplantation [1]. The Aspergillus species are leading fungal pathogens that can potentially cause aggressive forms of infection with a high mortality rate (up to 78%) [2]. The highest risk of infection in solid organ recipients is observed within the first 6 postoperative months due to optimal immunosuppression which impairs response to infectious pathogens’ passage into the lungs [3].

Despite the infection risk, anti-fungal prophylaxis is controversial and its use varies between transplant centers. The overall incidence of infective aspergillosis (IA) and other mouth infections are relatively low, the timing of occurrence is variable, and efficacy in prevention is uncertain, especially in terms of the risk of breakthrough invasive mouth infections with reduced susceptibility to the available antifungal therapy and drug-to-drug interactions [4]. Inhaling 24 mg of aerosolized Amphotericin B lipid complex every second day during the first week, and then once a week until discharge, may be beneficial [3,4]. Its use, however, involves a higher risk of drug-to-drug interference, electrolite imbalance, and kidney toxicity which is particularly important in the very early stage after heart transplantation.

The increased risk for opportunistic infection in solid organ recipients is predominantly related to lymphocyte deficiency. Immunosuppressive regimens have a lymphocyte-reducing effect with a secondary increase in the neutrophil-to-lymphocyte ratio (NLR), a marker of poor prognosis for the recipients [5,6].

The aim of the study was to evaluate the utility of NLR as a prognostic marker of survival and response to treatment of infective aspergillosis (IA) among patients after heart transplantation.

## 2. Materials and Methods

### 2.1. Patients

This retrospective, single-center, medium-volume study involved 15 heart recipients (13 (87%) male and 2 (13%) female) with the mean age of 55 ± 6 years, diagnosed with early invasive aspergillosis (IA) based on the EORTC/MSG group’s criteria between 2013 and 2019 [7]. The diagnosis was based on a diagnostic triad including clinical symptoms, laboratory findings (serum galactomannan slevels), and diagnostic test results (computed tomography (CT) and bronchofiberoscopy).

Patients were divided into two subgroups in terms of survival. Ten of them survived (67%) while the other five died, and IA was considered as the predominant reason. We retrospectively analysed peri-infective immunosupressive treatment, IA prophylaxis and therapy, and outcomes in the two groups. Data regarding clinical symptoms and laboratory and imaging exam results were collected. Whole blood samples for laboratory analysis were collected before the start of the antifungal therapy and compared with ones obtained on day three and day seven of treatment.

### 2.2. Study Approval

The study was conducted according to the guidelines of the Declaration of Helsinki and approved by the Institutional Ethics Committee of Poznan University of Medical Sciences, Poland (protocol No 246/21, date of approval 10/3/2021).

### 2.3. Statistical Analysis

Continuous variables were reported as mean ± standard deviation (SD) in case the data followed normal distribution (Shapiro–Wilk test). Otherwise, data were presented as median with the interquartile range Me [Q_1_–Q_3_]. The comparison of analyzed groups were compared using unpaired Student’s *t*-test or the Mann–Whitney test, depending on normality of the distribution. Categorical variables were reported as frequencies and percentages and compared using tests for proportions. The receiver operating characteristic (ROC) curve was used to find potential predictors of mortality. The optimal cut-off values were presented according to the highest sensitivity and specificity denoted by Youden’s index.

## 3. Results

### 3.1. Demographical and Clinical Analysis

Five out of fifteen patients diagnosed with IA died. The mean time from diagnosis to death in the deceased group was 28 ± 18 days. Demographic and clinical comparison of survivors and non-survivors, including results of diagnostic tools, is presented in Table 1.

The patients were diagnosed with IA in a mean time of 80 ± 53 (19–209) days after an orthotropic heart transplantation. The indication for the surgery was idiopathic dilated cardiomyopathy (DCM) in nine patients (60%) and end-stage heart failure secondary to ischemic heart disease (ICM, ischemic cardiomyopathy) in the remaining six patients (40%). All patients were tested for the risk of cytomegalovirus co-infection by its early antigen pp65, and the results were negative. The pp65 antigen was routinely measured in heart recipients following surgery during postoperative hospitalization and at the outpatient clinic. Standard prophylactic therapy included valganciclovir for three postoperative months.

There were two late deaths in the survival group (stroke and bacterial infection). Moreover, four rejection episodes were observed in the long-term follow-up after IA treatment in this group (mean follow-up of 5 (3–9) years).

### 3.2. Immunosuppressive Treatment

Immunosuppressive regimens involved a standard triple therapy: calcineurin inhibitor (tacrolimus) followed by antiproliferative drugs (mycophenolate mofetil) in all patients, and steroids in 14 patients (93%). Additionally, induction therapy (basiliximab) was applied in six (40%) patients due to a preoperative kidney failure. We did not find any survival differences in patients treated with basiliximab.

Methylprednisolone was given intravenously at a high dose of 125 mg three times daily for two postoperative days, followed by oral prednisolone at the dose of 1 mg per kg. The prednisolone dose would be lowered by 5 mg every third day to 20–40 mg as a supportive therapy.

The serum tacrolimus C0 levels for the first 3 postoperative months were 12 ± 7 ng/dL and were lowered after IA diagnosis. There were no episodes of rejections noted throughout the early study period until IA diagnosis.

Patients were on triple therapy including tacrolimus, mycophenolate mophetil, and steroids. The tacrolimus daily dosages were comparable between the 2 groups with 19.3 [13.4–21.6] vs. 17.4 [11.7–27.3] (*p* = 0.9025) and 17.6 [14.4–19.6] vs. 11.9 [11.1–14.9] (*p* = 0.0233) in the survivors and the deceased group, respectively.

The mycophenolate mophetil serum C-0 levels on admission were 2.1 [1.8–2.6] vs. 2.2 [1.9–3.0] (*p* = 0.9384); the steroids oral dose median values were 35 (30–40) mg vs. 35 (30–40 mg) (*p* = 1.000), respectively. The steroids were withdrawn as the IA diagnosis were made.

Following the heart transplant, all the patients received fluconazole as a prophylaxis of fungal infection, which is a standard protocol in our department. The fluconazole was given at the dose of 200 mg intravenously for 5 postoperative days, followed by 100 mg of oral therapy for the first postoperative year, as a standard preventive strategy. Amphotericin B was not used in the prevention due to the high risk of side effects.

### 3.3. Infective Aspergillosis Diagnosis and Treatment

The diagnosis of IA was based on a diagnostic triad composed of clinical symptoms, laboratory findings (serum galactomannan levels), and diagnostics results of CT and bronchofiberoscopy, with bronchoalveolar (BAL) lavage combined with a galactomannan test. The symptoms of a lower respiratory tract infection were followed by serum galactomannan levels measured by sensitivity-specificity of galactomannan-enzyme immunoassay (GM-EIA) with mean values of 4.7 ± 3.7. CT scan results were evaluated in terms of involvement of the lung segments.

Detailed information is presented in Table 1. Aspergillosis infection involved only lung tissue. The detailed information regarding hospitalization are presented in Table 2.

Upon confirmation of the diagnosis, therapy including voriconazole with concomitant micamine and caspofungine was commenced. Under voriconazole therapy, the tacrolimus daily dosages were lowered according to serum levels.

Detailed findings of the blood morphology analysis are presented in Table 3. Day three of therapy was chosen as a standard response time after antifungal therapy commencement. Day seven was chosen because deterioration was noted after one week of unsuccessful therapy in the deceased group.

### 3.4. Laboratory Results, Clinical Symptoms and CT Scan Analysis

A significant difference in the neutrophil-to-lymphocyte ratio (NLR) calculated before therapy initiation was found between the two groups, with median [Q1–Q3] values of 7.4 (6.2–18.9) in survivors and 30.8 (10.2–36.1) in the deceased group.

During the initial 7 days of therapy, the NLR significantly differed between the two groups on day 3 and day 7 with median values of 10.8 [4.3–17.0] vs. 20.2 [17.4–116.8] (*p* = 0.0373) and 5.2 [3.2–8.1] vs. 32.2 [13.5–49.9] (*p* = 0.0101) in the survivor and the deceased group, respectively. We followed the results of NLR prior to heart transplantation and received the results of 3.6 [2.8–4.3] vs. 2.7 [2.5–11.7] (*p* = 0.7133). The individual presentation of NLR changes in each patient is presented in Table 4.

The NLR was a significant predictor of death both at day three and day seven of therapy. The cut-off point was 17.02 for day 3, with a sensitivity of 80% and specificity of 80%, and 12.08 for day 7, with a sensitivity of 90% and specificity of 80%. The ROC curves analysis is presented in Figure 1 and Figure 2.

Although the sample size is very low, the logistic regression analysis shows that NLR, after seven days of therapy, predicts patients’ death. Patients with NLR after 7 days of therapy above 12.08 were 16 times more likely to die (OR = 16, 95%CI (1.09–243.2), *p* = 0.0429). 

Leukocyte count (7.7 ± 5 vs. 8 ± 5 10 e/L; 7.5 ± 4 vs. 10.6 ± 7 10 e/L; 6 ± 2 vs. 11.7 ± 10 e/L) and neutrophil count (6.6 ± 4.6 vs. 7.2 ± 4.8 10 e/L; 6.3 ± 3.9 vs. 10 ± 6 10 e/L; 4.7 ± 1.5 vs. 11 ± 7.5 10 e/L) were analyzed consecutively—prior to the initiation of antifungal therapy, on day three, and day seven of the therapy. The results, however, did not differ significantly between the two groups in the Cox analysis (Table 3).

During the antifungal therapy, serum immunosuppressive drug levels (C-0) were stable prior to the antifungal therapy as well as on day 3 and day 7, including calcineurin inhibitor (tacrolimus) values of 17.7 [15.6–20.2] vs. 17.4 [12.4–25.6] (*p* = 0.9024); 19.3 [13.4–21.6] vs. 17.4 [11.7–27.3] (*p* = 0.9025); 17.6 [14.4–19.6] vs. 11.9 [11.1–14.9] (*p* = 0.0233) in the survivors and the deceased group, respectively. Serum mycophenolate mophetil level (C-0) was measured prior to initiation of the therapy and on day 3 and day 7 of the therapy, with values of 2.1 [1.8–2.6] vs. 2.2 [1.9–3.0] (*p* = 0.9384); 2.5 [1.3–3.4] vs. 2.3 [1.9–2.6] (*p* = 0.9322); 3.2 [2.5–3.7] vs. 2.7 [1.9–3.3] (*p* = 0.4391), respectively.

The symptoms and segmental lung involvement did not differ between the two groups (Table 3). The CT findings (Table 3) indicate a more advanced form of mold infection in the deceased group with a high incidence of large nodules than in the survivor group—5 patients (100%) vs. 3 patients (30%), respectively. The “halo” sign was more often found in the deceased group (4 patients (80%)) than in the survivor group (3 patients (30%)).

CTs and chest X-rays in patients from the survivor and deceased groups are presented in Photos 1–4. A chest X-ray in a survivor (Figure 3a) and a deceased (Figure 3b) patient are shown. Inflammatory features involve the whole left lung in Figure 3a and the middle and upper right lobes of the right lung in Figure 3b.

Figure 4a presents pulmonary tissue infiltrations in a survivor patient and large nodules are presented in a deceased patient in Figure 4b.

## 4. Discussion

Our study presents the results of an NLR comparison between surviving and deceased heart recipients suffering from early IA infections. To the best of our knowledge, this is the first study indicating that a simple parameter such as NLR may predict mortality risk for mold infection following heart transplantation. The combination of leukocyte-reducing effects with secondarily increased neutrophils in peripheral blood presented as NLR in immunocompromised patients has already been postulated [4]. The results of the study confirm negative implications of hosting defense and inflammatory suppression in immunocompromised patients for their survival of IA. More interestingly, preoperative NLR results were not statistically different between the two groups, and, conversely, were significantly different in the onset of IA.

This study reveals a significant correlation between heart recipients’ lymphocyte depression secondary to antirejection therapy in heart recipients and survival. Postoperatively, no episodes of rejection were observed in our group. Methylprednisolone was administered intravenously at a high dose of 125 mg three times daily for 2 postoperative days, followed by oral prednisolone at the dose of 1 mg per kg. The prednisolone dosage would be lowered by 5 mg every third day to 20–40 mg as a supportive therapy.

The mechanism underlying the association between high NLR and a poorer outcome could be due to the association of NLR representing the depressed systemic inflammatory response as presented in tumor studies [8,9]. The negative correlation between preoperative assessment of the NLR status and tumor progression and prognosis in patients with oncologic disease has already been observed [9].

Patients with high NLR on antifungal therapy had a poorer prognosis of IA survival. On immunosuppressive therapy, the lymphocytes are depressed, which is evidenced by high NLR values, leading to suppression of the natural defense system, including inflammatory response. According to the results of our study, a combination of a targeted antifungal therapy and decreased NLR values which represent the body’s self-defense mechanism is essential for the patient’s survival.

Serum immunosuppressive drug levels were stable throughout the antifungal therapy, although the drug dosages were adjusted due to the interaction between antirejection and antifugal drugs interaction. Based on our results, we suggest that NLR should be a warrant for immunosuppression dosage modification to achieve systemic immune response to infection. Those patients who present with high NLR on day three and day seven of the therapy should be considered for immunosuppression therapy modification. We believe that a peripheral marker (NLR) may be useful in everyday practice.

Aspergillosis is one of the most common opportunistic infections following heart transplantation, with relatively high mortality reaching 14–30% [10]. The diagnosis of the infection is based on such criteria as serum galactomannan levels over 0.5 [11] combined with radiographic diagnostic criteria and bronchoscopy results [12]. We found that survivors presented with less extensive lung involvement in X-ray imaging. BAL results that are characterized with specificity as high as 96% had high levels in our study, with mean values of 5.1 ± 3.1 in the survivors and 4.1 ± 3.8 in the deceased group, respectively (*p* = 0.678) [13].

All study subjects were treated with triple immunosuppressive therapy including tacrolimus, mycophenolate mophetil, and steroids. The strong correlation between aspergillosis infection in immunocompromised patients and corticosteroids has already been proven by Gustafson [14]. NLR was also found to be an easy and reliable parameter of the first year mortality following heart transplantation [15]. High NLR may be caused by low absolute lymphocyte count and result in inadequate prevention of opportunistic infections [16]. The concept of low lymphocyte count indicating insufficient reserves to mount an appropriate response to pathogens may explain the increased mortality rate. Moreover, reduced neutrophil recruitment following heart transplantation has been postulated by Li as a significant factor for decreased immunological self-defense of solid organ recipients [17]. In our study, the observed decreasing values of NLR in consecutive days of aspergillosis therapy, were found to be a predictor of survival. The decreasing NLR on both day three and day seven of the therapy were significant predictors of survival. The cut-off point for day 3 following therapy initiation was <17.02 with a sensitivity of 80% and specificity of 80%. The cut-off point for day 7 following therapy initiation was <12.08 with a sensitivity of 90% and specificity of 80%. Decreasing values of a marker as simple as NLR within the first week of the therapy present a high predictive value for recovery. Our study confirmed that the more the immunologic system is suppressed, as evidenced by high NLR, the higher the risk of ominous results of IA infection.

Group 2 (the deceased group) was characterized by statistically significant lower values of lymphocytes and non-statistically significant higher values of WBC and neutrophils. This observation may be partly explained by the concomitant bacterial co-infection. The significantly higher mortality rates in invasive aspergillosis complicated by either bacterial or viral co-infection were presented in previous studies [18,19]. Asin et al., in their meta-analysis, showed the positive relationship between IA and posttransplant bacterial infection and respiratory tract viral infection [20]. The significantly lower values of lymphocytes highlight insufficient capability of immune system to manage infection, especially in complex microorganisms’ involvement [21].

Patients with higher NLR are more immunosuppressed and, therefore, are at an increased risk of dying from severe infections such as IA. NLR can be regarded as novel additive parameter of infection risk in immunocompramised patients irrespectively of serum antirejection drug levels.

### Study Limitation

This is a single-center retrospective study with a limited number of study subjects. Consequently, comparisons by means of statistical methods may be subject to bias. We cannot exclude the possibility that the inclusion of a greater number of patients would have increased the power of our statistical analysis and could have yielded more interesting findings. A detailed analysis of very basic blood morphology parameters and their potential prognostic value, however, warrant further research with more subjects included.

## 5. Conclusions

The initial findings in our study indicate that NLR, as a simple parameter of very basic blood analysis, may be of predictive value in the estimation of mortality risk or response to treatment among patients with invasive aspergillosis following heart transplantation.

## Figures and Tables

**Figure 1 medicina-57-01300-f001:**
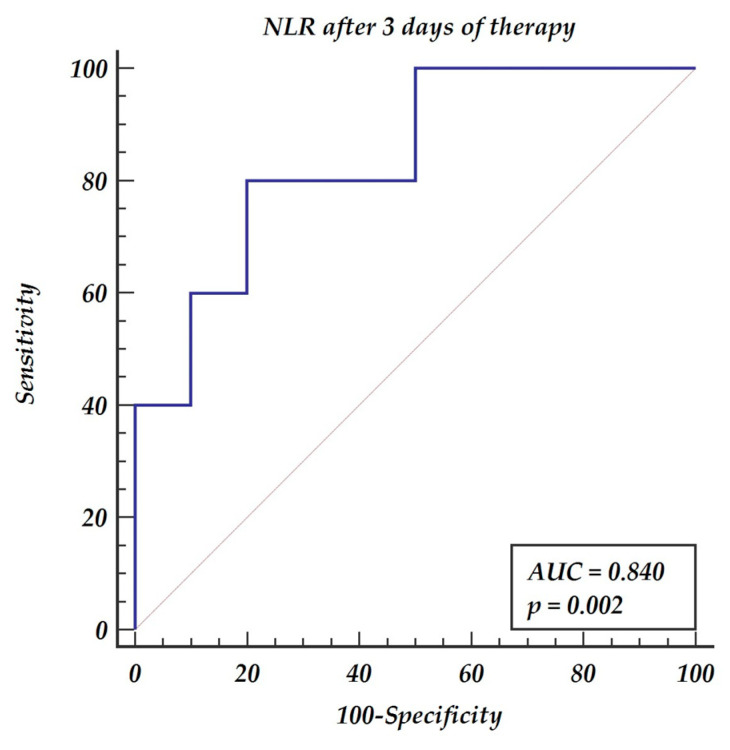
Receiver operating characteristic (ROC) curves for the neutrophil-to-lymphocyte ratio after 3 days of antimold therapy.

**Figure 2 medicina-57-01300-f002:**
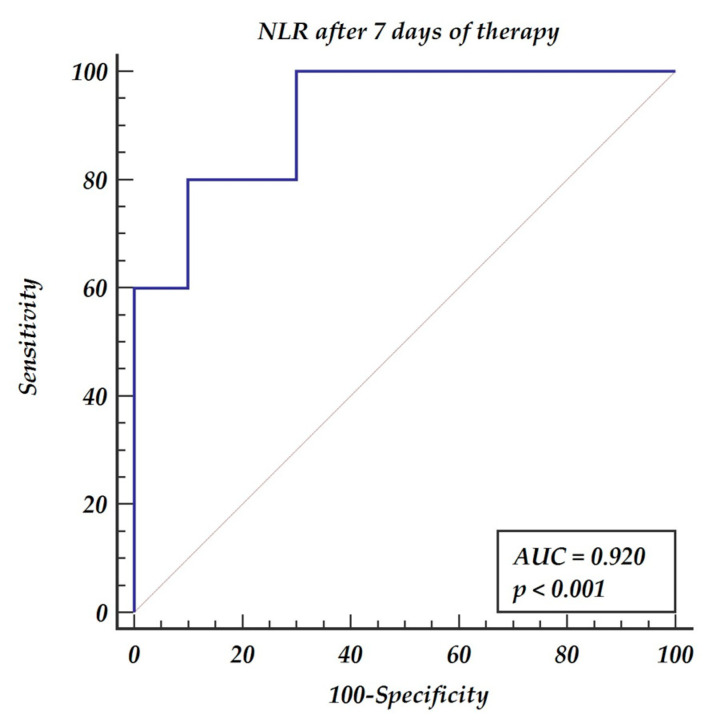
Receiver operating characteristic (ROC) curves for the neutrophil-to-lymphocyte ratio after 7 days of antimold therapy.

**Figure 3 medicina-57-01300-f003:**
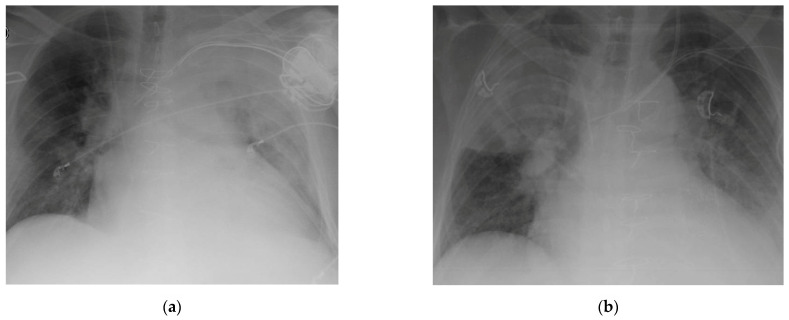
(**a**) Survival group. IA Infiltration of left lung. (**b**). Deasesed group. Right upper and middle lobe involved.

**Figure 4 medicina-57-01300-f004:**
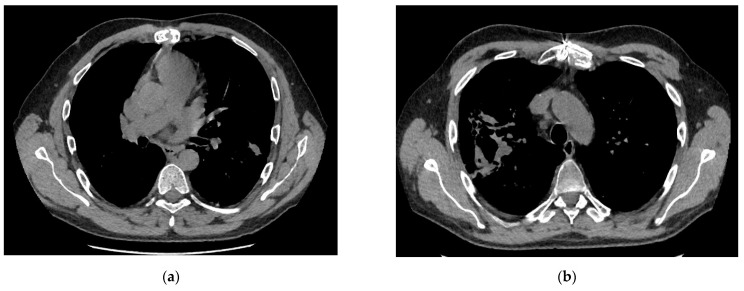
(**a**) Survival group. IA nodules in the left lung. (**b**) Deasesed group. Right middle lobe involvement with cavities and left with small nodules.

**Table 1 medicina-57-01300-t001:** Demographic and clinical characteristics of IA subjects.

	Group 1 Survival (*n* = 10)	Group 2 Deceased (*n* = 5)	*p*-Value
Gender (M/F)	9 (90%)/1 (10%)	4 (80%)/1 (20%)	0.591
Age	56 ± 5	54 ± 9.3	0.591
Initial diagnosis (DCM/ICM)	6 (60%)/4 (40%)	3 (60%)/2 (40%)	1.000
Co-morbidities:			
1. Arterial hypertension	4 (40%)	2 (40%)	0.895
2. Diabetes mellitus	3 (30%)	2 (40%)	0.689
3. Kidney dysfunction	5 (50%)	2 (40%)	0.867
4. Hipercholesterolemia	4 (40%)	2 (40%)	0.919
5. Atrial fibrillation	1 (10%)	1 (20%)	0.578
6. Stroke	2 (20%)	1 (20%)	0.878
7. Chronic obstructive pulmonary disease	2 (20%)	1 (20%)	0.578
Aspergillosis:			
Onset (days following HTX)	79 ± 43	83 ± 75	0.896
Symptoms:			
Fever	4 (40%)	3 (60%)	0.705
Cough	8 (80%)	4 (80%)	0.121
No symptoms	1 (10%)	1 (20%)	1.000
Fatigue	7 (70%)	4 (80%)	0.264
Shortness of breath	4 (40%)	2 (40%)	0.439
Laboratory:			
Serum galactomannan level	5 ± 3.9	4.1 ± 3.8	0.678
CT scan:			
Segment involved	3.5 [2–7]	2 [2–7.25]	0.960
Left lobe	1.5 [1–4]	2 [1.5–2.25]	0.736
Right lobe	2.5 [2–3]	2 [0–4.5]	0.904
Findings:			
Consolidations	6	2	0.464
Nodules	4	6	
Small	1	1	0.7469
Large	3	5	0.747
Parabronchial infiltrate	1	1	0.591
Crescent sign	3	4	0.067
Bronchoscopy:	7 (70%)	5 (100%)	
BAL galactomannan	5.2 ± 2.1	5.3 ± 2.3	0.934

Continuous variables are presented as either means ± standard deviations (sd) (if normally distributed) or medians with interquartile ranges, while categorical data are presented as numbers (*n*) with percentages (%). Abbreviations: BAL—bronchoalveolar lavage, CT—computed tomography, DCM—dilated cardiomyopathy, F—female, IA—invasive aspergillosis, ICM—ischemic cardiomyopathy, M—male.

**Table 2 medicina-57-01300-t002:** Hospitalization details.

	Group 1 Survival (*n* = 10)	Group 2 Deceased (*n* = 5)	*p*-Value
Hospitalization:			
1. Overall (days)	21 (14–29)	31 (6–77)	*p* = 0.056
2. ICU (days)	2 (1–14)	13 (9–25)	*p* = 0.008 *
3. Mechanical ventilation (pts)	3 (30%)	5 (100%)	*p* = 0.044 *
- hospitalization day when mechanical support was required	10 (8–12)	11 (9–18)	*p* = 0.786
- mechnical ventilation therapy (hrs)	40 (24–240)	308 (222–583)	*p* = 0.001 *
4. Amin pressors requirements (pts)	0	3 (60%)	
Concomitant bacterial infection:	1 (10%)	4 (80%)	*p* = 0.034 *
- onset of diagnosis regarding hospitalization (days)	11 (9–12)	11 (8–14)	*p* = 0.934
- pathogen	*Escherichia coli*-1	*Escherichia coli*-2	
		*Proteus mirabilis*-2	
- therapy	Meropenemum	Meropenemum	

Abbreviations: ICU—intensive care unit, hrs—hours, pts—patients, *—statistically significant.

**Table 3 medicina-57-01300-t003:** Laboratory results before, on day three, and day seven of the therapy.

	Group 1, No = 10	Group 2, No = 5	*p*-Value
Before therapy			
1. WBC × 10^9^/L	7.77 ± 4.9	8.0 ± 5.1	0.919
2. Neutrophils × 10^9^/L	6.7 ± 3.9	7.2 ± 4.9	0.835
3. Lymphocytes × 10^9^/L	0.64 ± 0.3	0.3 ± 0.1	0.049
4. NLR	3.6 [2.8–4.3]	2.7 [2.5–11.7]	0.178
5. RDW (%)	17.2 ± 2	15.9 ± 0.3	0.141
6. Plt × 10^3^/L	249 ± 54	232 ± 142	0.807
7. PDW (fL)	55.5 ± 5.1	63 ± 5.1	0.037
After 3 days of therapy			
1. WBC × 10^9^/L	7.4 ± 4	10.7 ± 6.5	0.462
2. Neutrophils × 10^9^/L	6.3 ± 3.9	10 ± 6.4	0.391
3. Lymphocytes × 10^9^/L	0.65 ± 0.4	0.3 ± 0.14	0.014 *
4. NLR	10.8 [4.3–17.0]	20.2 [17.4–116.8]	0.037 *
5. RDW (%)	17.4 ± 2.1	16.1 ± 0.7	0.244
6. Plt × 10^3^/L	248 ± 77	220 ± 100	0.807
7. PDW (fL)	58.2 ± 11	64.7 ± 5.8	0.327
After 7 days of therapy			
1. WBC × 10^9^/L	6.1 ± 1.9	11.7 ± 7.8	0.111
2. Neutrophils × 10^9^/L	4.8 ± 1.5	11.05 ± 7.5	0.066
3. Lymphocytes × 10^9^/L	0.95 ± 0.6	0.36 ± 0.2	0.037 *
4. NLR	5.2 [3.2–8.1]	32.2 [13.5–49.9]	0.010 *
5. RDW (%)	17.2 ± 1.9	16.3 ± 0.7	0.387
6. Plt × 10^3^/L	243 ± 68	243 ± 140	0.807
7. PDW (fL)	58 ± 12	63 ± 7	0.327

Continuous variables are presented as either means ± standard deviations (SD) (if normally distributed) or medians with interquartile ranges. Abbreviations: NLR—neutrophil to lymphocyte ratio, PDW—platelet distribution width, Plt—platelets, RDW—red blood cell distribution, WBC—white blood count, *—statistically significant.

**Table 4 medicina-57-01300-t004:** NLR and their changes in particular patients.

Patients	NLR on Admission	Median Difference on Admission/3rd Day	NLR 3rd Day	Median Difference on Admission/7th day	NLR 7th Day
1	31.6	126.2	157.8	14.3	45.9
2	6.1	14.2	20.3	−1.2	4.9
3	8.1	4.6	12.7	−2.6	5.5
4	21.1	−4	17.1	9.3	30.4
5	12.8	−9.6	3.2	−10.1	2.7
6	38.8	−26	12.8	−26.7	12.1
7	30.8	72.4	103.2	31.3	62.1
8	5.4	6.5	11.9	26.8	32.2
9	4.2	0.1	4.3	−0.8	3.4
10	6.6	0.5	7.1	−3.3	3.3
11	11.8	8.4	20.2	3.7	15.5
12	4.8	−1.8	3	−2.4	2.4
13	6.7	2.2	8.9	1.5	8.2
14	18.9	0.5	19.4	−12.1	6.8
15	49.3	−30	19.3	−41.7	7.6

Abbreviations: NLR—neutrophil to lymphocyte ratio.

## Data Availability

All data will be available for three years after obtaining a reasonable request.

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
