# Peer review of "Neutrophile-to-Lymphocyte Ratio as a Predictor of Mortality and Response to Treatment in Invasive Aspergillosis among Heart Transplant Recipients—Exploratory Study"

_medicina, 2021, doi:10.3390/medicina57121300_

Round 1
Reviewer 1 Report
In this study, Tomasz Urbanowicz et al. report on the neutrophile-to-lymphocyte ratio as a predictor of mortality and response to treatment in invasive aspergillosis among heart transplant recipients. This is an interesting manuscript and an important report for determining the predictor of survival after aspergilosis in heart transplantation recipients. In the past literature, there is little insight into this issue. This report focuses on that point and is worth reporting.
The biggest problem of this study is statistically weakness because the number of cases is too small, as the author has shown in the limitation. It seems better to report as a case report than as a clinical trial. Therefore, clarifying more detail information about the characteristics of the two groups will increase the value of this study. Please consider the following points.
There is too little information of patients' characteristics in this study. Please show comorbidities such as hypertension, hyperlipidemia, diabetes, CKD and so on. Also, please describe detail information on immunosuppressive drugs ( if possible, including blood concentration) in each patients, especially the amount of steroid. Also, please show treatment information that may affect the clinical course during hospitalization (use of catecholamine and ventilators, other drugs). Did you administer any drugs that affect the blood concentration of immunosuppressive drugs?
Please clarify detail information about the 5 dead cases as much as possible. Length of hospitalization to death, period of oxygen administration, length of stay in the intensive care unit, etc.
It would be very interesting if it were possible to evaluate the prognosis of post-transplant Aspergillus infection with simple parameter such as NLR. However, there are statistical limitations in analyzing NLR alone as a single predictor of mortality. What is preferable for clinical research, the number of cases should be increased and multivariate analysis should be performed to evaluate the ROC of all strong factors. As you pointed out in the discussion, it is also an important question whether improving NLR will result in improved outcomes. Have you had any of the cases improve NLR? Please show changes of NLR in individual cases rather than mean values, if possible in All 15 cases.
In addition, were there any factors that increased white blood cells relatively affected NLR, such as bacterial infection or bacterial pneumonia, amount of steroid, smoking, etc ?
Please include images of chest x ray and chest CT in two typical patients (improved and aggravated) in your article. It will increase the educational value of this article.
Author Response
Poznan, Poland, 17th November 2021
Dear Reviewer,
in response to your valuable suggestions we enclosed the following modifications answering to your suggestions point by point:
The biggest problem of this study is statistically weakness because the number of cases is too small, as the author has shown in the limitation. It seems better to report as a case report than as a clinical trial. Therefore, clarifying more detail information about the characteristics of the two groups will increase the value of this study. Please consider the following points.
- There is too little information of patients' characteristics in this study. Please show comorbidities such as hypertension, hyperlipidemia, diabetes, CKD and so on. Also, please describe detail information on immunosuppressive drugs ( if possible, including blood concentration) in each patients, especially the amount of steroid. Also, please show treatment information that may affect the clinical course during hospitalization (use of catecholamine and ventilators, other drugs). Did you administer any drugs that affect the blood concentration of immunosuppressive drugs?
Ad 1a. comorbidities were included in Table 1.
Ad 1b. immunosuppression – lines 131 – 137 were added
Ad 1c. catecholamines, ventilators – were enclosed in Table 2
- Please clarify detail information about the 5 dead cases as much as possible. Length of hospitalization to death, period of oxygen administration, length of stay in the intensive care unit, etc.
Regarding 5 cases:
Ad 2a. Length of hospitalization – TABLE 2 was included
Ad 2b. mechanical ventilation – TABLE 2 was included
Ad 2c. length of stay on ICU – TABLE 2 was included
- It would be very interesting if it were possible to evaluate the prognosis of post-transplant Aspergillus infection with simple parameter such as NLR. However, there are statistical limitations in analyzing NLR alone as a single predictor of mortality. What is preferable for clinical research, the number of cases should be increased and multivariate analysis should be performed to evaluate the ROC of all strong factors. As you pointed out in the discussion, it is also an important question whether improving NLR will result in improved outcomes. Have you had any of the cases improve NLR? Please show changes of NLR in individual cases rather than mean values, if possible in All 15 cases.
Ad 3. NLR changes in each pts enclosed into the study presented in TABLE 4
In addition, were there any factors that increased white blood cells relatively affected NLR, such as bacterial infection or bacterial pneumonia, amount of steroid, smoking, etc ?
Ad 4. Was included in text
Please include images of chest x ray and chest CT in two typical patients (improved and aggravated) in your article. It will increase the educational value of this article.
Ad 5. Photos 1-4 and additional text in lines 225-241 were added
Kind regards
Tomasz Urbanowicz

Reviewer 2 Report
this st
Dr. Tomasz Urbanowicz and colleagues have presented a study on NLR as predictor factor of mortality and response to treatment in invasive aspergillosis among 15 heart transplantation patients.
Though this appears to be the first report of such an association in these patients that would be helpful to the readers, the following revisions are suggested:
1) In the result section, beside reporting the p value, please mention the effect size of this association. For example using mean difference(MD) or stanardised mean difference(SMD) or hazard risk(HR). It would help to demonstrate the size of association.
2) According to low sample size and low powered study, it is recommended to add “exploratory study” at the end of title.
Author Response
Poznan, 17th November 2021
Dear Reviewer,
in response to your valuable suggestions we enclosed the following modifications answering to your suggestions:
Though this appears to be the first report of such an association in these patients that would be helpful to the readers, the following revisions are suggested:
1) In the result section, beside reporting the p value, please mention the effect size of this association. For example using mean difference(MD) or stanardised mean difference(SMD) or hazard risk(HR). It would help to demonstrate the size of association.
We enclosed the corrections in text
2) According to low sample size and low powered study, it is recommended to add “exploratory study” at the end of title.
Ad 2.
“Neutrophile-to-lymphocyte ratio as a predictor of mortality and response to treatment in invasive aspergillosis among heart transplant recipients – exploratory study.”
Kind regards
Tomasz Urbanowicz

Round 2
Reviewer 1 Report
Author Tomasz Urbanowicz et al. is to be congratulated for a nicely improved your article. The author almost addressed my concerns sufficiently in this version. However, the following a point need to be corrected.
As I directed, you provided detailed information about the patient. According to Table 2, 80% of Group 2 was complicated by a bacterial infection. Group 2 clearly has lower lymphocytes than Group 1 in table 3, but the increasing of WBC may be due to a bacterial infection.
If it is possible that NLRs have increased relatively due to factors that increase WBC, some considerations need to be added in discussion.
Author Response
Poznan, Poland, 21st November 2021
Dear Reviewer,
thank you, once again, for your valuable suggestions. We enclosed the following modifications answering to your comments and marked them using red color. We also added four more references:
The Group 2 (deceased) was characterized by statistically significant lower values of lymphocytes and non-statistically significant higher values of WBC and neutrophils. This observation may be partly explained by the fact of concomitant bacterial co-infection. The significantly higher mortality rates in invasive aspergillosis complicated by either bacterial or viral co-infection were presented in previous studies [18.19]. Asin et al. in their meta-analysis showed the positive relationship between IA and posttransplant bacterial infection and respiratory tract viral infection [20]. The significantly lower values of lymphocytes highlight insufficient capability of immune system to manage infection, especially in complex microorganisms’ involvement [21].
Burghi G, Lemiale V, Seguin A, Lambert J, Lacroix C, Canet E, Moreau AS, Ribaud P, Schnell D, Mariotte E, Schlemmer B, Azoulay E. Outcomes of mechanically ventilated hematology patients with invasive pulmonary aspergillosis. Intensive Care Med. 2011; 37: 1605-12.
Schauwvlieghe AFAD, Rijnders BJA, Philips N, Verwijs R, Vanderbeke L, Van Tienen C, Lagrou K, Verweij PE, Van de Veerdonk FL, Gommers D, Spronk P, Bergmans DCJJ, Hoedemaekers A, Andrinopoulou ER, van den Berg CHSB, Juffermans NP, Hodiamont CJ, Vonk AG, Depuydt P, Boelens J, Wauters J; Dutch-Belgian Mycosis study group. Invasive aspergillosis in patients admitted to the intensive care unit with severe influenza: a retrospective cohort study. Lancet Respir Med. 2018; 6: 782-792.
Pérez-Jacoiste Asín MA, López-Medrano F, Fernández-Ruiz M, Silva JT, San Juan R, Kontoyiannis DP, Aguado JM. Risk factors for the development of invasive aspergillosis after kidney transplantation: Systematic review and meta-analysis. Am J Transplant. 2021; 21: 703-716.
Schroth J, Weber V, Jones TF, Del Arroyo AG, Henson SM, Ackland GL. Preoperative lymphopaenia, mortality, and morbidity after elective surgery: systematic review and meta-analysis. Br J Anaesth. 2021 Jul;127(1):32-40.
Kind regards
Tomasz Urbanowicz
on behalf of all co-authors
